# Mechanical Properties of Cubene Crystals

**DOI:** 10.3390/ma15144871

**Published:** 2022-07-13

**Authors:** Leysan Kh. Galiakhmetova, Igor S. Pavlov, Ayrat M. Bayazitov, Igor V. Kosarev, Sergey V. Dmitriev

**Affiliations:** 1Institute for Metals Superplasticity Problems of RAS, 450001 Ufa, Russia; rysaeva.l.h@gmail.com; 2Mechanical Engineering Research Institute of the Russian Academy of Sciences—Branch of Federal Research Center “Institute of Applied Physics of RAS”, 603024 Nizhny Novgorod, Russia; ispavlov@mail.ru (I.S.P.); ayrseptember@mail.ru (A.M.B.); 3Institute of Molecule and Crystal Physics, UFRC of RAS, 450075 Ufa, Russia; igork23v@mail.ru

**Keywords:** cubene, elastic constants, equilibrium phases, fullerite, molecular dynamics

## Abstract

The fullerene family, whose most popular members are the spherical C${}_{60}$ and C${}_{70}$ molecules, has recently added a new member, the cube-shaped carbon molecule C${}_{8}$ called a cubene. A molecular crystal based on fullerenes is called fullerite. In this work, based on relaxational molecular dynamics, two fullerites based on cubenes are described for the first time, one of which belongs to the cubic system, and the other to the triclinic system. Potential energy per atom, elastic constants, and mechanical stress components are calculated as functions of lattice strain. It has been established that the cubic cubene crystal is metastable, while the triclinic crystal is presumably the crystalline phase in the ground state (the potential energies per atom for these two structures are −0.0452 and −0.0480 eV, respectively).The cubic phase has a lower density than the monoclinic one (volumes per cubene are 101 and 97.7 Å${}^{3}$). The elastic constants for the monoclinic phase are approximately 4% higher than those for the cubic phase. The presented results are the first step in studying the physical and mechanical properties of C${}_{8}$ fullerite, which may have potential for hydrogen storage and other applications. In the future, the influence of temperature on the properties of cubenes will be analyzed.

## 1. Introduction

The existence of large carbon molecules was predicted and described in some early papers, but a paper [1] published in 1985 caused an avalanche of publications on fullerenes. The smallest spherical fullerene is C${}_{20}$; the most common molecules are C${}_{60}$ and C${}_{70}$, but fullerenes with a larger number of carbon atoms are easily obtained [2,3]. Probably the smallest possible carbon cluster is a cubic C${}_{8}$ molecule called cubene [4].

Cubenes C${}_{8}$ and cubanes C${}_{8}$H${}_{8}$ [5,6] are less studied than C${}_{60}$ and C${}_{70}$ fullerenes. The cubene molecule is highly strained, since a bond angle of 90${}^{\circ }$ is energetically unfavorable; however, as follows from the ab initio study [7], the molecule should be stable because its energy corresponds to a deep minimum. Structural, electronic, and optical properties of cubane C${}_{8}$H${}_{8}$ and cubanoids (cubane-like molecules with hydrogen substituted by other atoms) were analyzed using a DFT approach [8].

Derivatives of cubane such as octanitrocubane and heptanitrocubane are excellent candidates for storage of large amounts of energy [9]. Thermochemical calculations show that 4145 kJ/mol is released during decomposition of the first, and 4346 kJ/mol of the second. Cuban can be used to obtain various bioisosteres, i.e., compounds obtained by replacing a group of atoms with a cubane, while maintaining the biological activity of the original compound. Thus, new substances with improved properties compared to the parent compound for pharmaceutical and agrochemical applications can be created [10]. Cubanes are also promising for hydrogen storage technologies [11,12,13].

Covalently bonded arrays of cubanes were shown to be possible [14]. Chemical modification of cubanes and cubane oligomers was analyzed in the work [15]. The electronic structure of C${}_{8}$ and B${}_{4}$N${}_{4}$ cubic systems was analyzed [16]. It was shown that the functionalization of a chain of covalently bonded cubanes leads to a significant change in its conductivity [17,18]. This means that the electronic properties of cubane-based oligomers can be tuned by functionalization for applications in nanoelectronics.

Molecule crystals composed of fullerenes are called fullerites; they possess many promising physical, mechanical, and chemical properties for applications. Polymerization of fullerites under high pressure and temperature produces extremely hard materials with bulk modulus from 454 GPa to 644 GPa [19]. Fullerite demonstrates an anomalously slow thermal relaxation due to the excitation of internal vibrational degrees of freedom [20].

The mechanical properties of molecular crystals may differ from those of elemental solids, since the rotational degrees of freedom of molecules may play an important role [21,22,23,24,25]. There exist many examples of molecular crystals where rotations of molecules were analyzed [26,27,28,29,30]. Such crystals support rotobreathers that are dynamical regimes, in which one molecule (or particle) rotates, while others oscillate [31,32,33,34,35]. Properties of graphene can be described using particles with rotational degrees of freedom [36,37].

Metamaterials with particles having rotational degrees of freedom (polygons or polyhedrons) can be created artificially [38,39,40,41,42]. The rotational degrees of freedom can produce negative thermal expansion [21,22,23,24,25] or auxetic behaviour (negative Poisson’s ratio) [38,39,40,41,42]. The crystal has been potentially used in several applications, such as bearing material for total hip implants [43].

The crystal structure of fullerites is of great importance, since the structure determines the properties of crystals. Fullerite C${}_{60}$ can be packed into a simple cubic or face-centered cubic (fcc) lattice [44,45]. In both lattices, a primitive translational cell contains one fullerene molecule. In the theoretical work [46], it was shown that, for C${}_{60}$, the fcc structure is more stable than hexagonal close-packing (hcp), but for C${}_{70}$ hcp has lower energy than fcc lattice. The fcc structure of C${}_{60}$ at 0 K has an orthorhombic distortion due to ordering of fullerenes. At higher temperatures, there is a transition to an orientationally disordered fcc lattice [46]. As can be seen, many studies have been carried out to predict and analyze the structure of fullerites C${}_{60}$ and C${}_{70}$, but we have not been able to find any information about the structure for fullerite C${}_{8}$ except for the study [47] where cubic cubane C${}_{8}$H${}_{8}$ was described. In the present study, we set ourselves the task of finding the equilibrium phases of a cubene crystal and their mechanical properties.

## 2. Materials and Methods

Molecular dynamics is used to simulate the structure and mechanical properties of cubene crystals. Carbon–carbon interactions are described by the harmonic potential, while non-valence interactions are described by the Lennard–Jones potential. Each cubene interacts with the nearest, second and third neighboring cubenes. Below is a detailed description of the model.

A model of cubene molecule is shown in Figure 1. Each vertex of a cube with side $\rho_{0}$ contains a carbon atom. The covalent bonding of atoms in cubenes makes them relatively rigid. Cubenes interact with each other through much weaker van der Waals bonds.

In the present work, the equilibrium phases of a cubene crystal are described, and, in this case, cubene deformation plays an insignificant role. Therefore, covalent bonds are described by a simple harmonic potential. It is assumed that each carbon atom interacts through covalent bonds with the three nearest atoms and with the most distant atom, for example, atom 1 interacts with atoms 2, 4, 5, and 7 (see Figure 1). The interaction between the farthest neighbors is introduced to ensure the stability of the cubic shape of the molecule.

The equilibrium phases of cubene crystal are found using perturbative relaxational molecular dynamics. In this approach, the positions of the atoms are first perturbed to drive the system away from a local potential minimum, and then the structure is relaxed to reach a deeper energy minimum. Initially, cubenes with random orientation are arranged in the fcc lattice with periodic boundary conditions. The simulations are carried out for a computational cell that includes 2×2×2 primitive translational cells of the fcc lattice. The total number of cubenes in the computational cell is M=8, and the total number of carbon atoms is 64. The structure relaxation is performed with respect to the coordinates of the carbon atoms and with respect to the vectors specifying the periodic cell of the structure in order to achieve the state with zero forces acting on the atoms and zero external stresses applied to the computational cell.

Our model is built in the spirit of the work [20], where the deformation of valence bonds is described by the harmonic potential, and intermolecular interactions by the Lennard–Jones potential. Following this work, the potential energy of the cubene crystal described by the periodic computational cell with *M* cubenes can be written as follows:(1)$$\begin{array}{c} P=\sum_{m=1}^{M}\sum_{n=1}^{12}U_{n}+\sum_{m=1}^{M}\sum_{k=1}^{4}V_{k}+\sum_{i>j}\phi (|R_{i,j}\left|\right). \end{array}$$

The first term on the right-hand side of Equation (Equation 1) describes the valence interactions between nearest atoms in the cubenes; there are 12 nearest bonds in each cubene. The harmonic potential is used:(2)$$\begin{array}{c} U_{n}=\frac{K}{2}\left(\right|r_{n}{\left|-\rho_{0}\right)}^{2}, \end{array}$$
where K=31.77 eV/Å${}^{2}$ is the bond stiffness [48], $|r_{n}|$ is the distance between two nearest neighbors in a cubene molecule and $\rho_{0}=1.418$ Å is the equilibrium bond length. From Figure 1, it can be seen that the twelve bonds between nearest neighbors connect the following pairs of atoms: 1-2, 2-3, 3-4, 4-1, 1-5, 2-6, 3-7, 4-8, 5-6, 6-7, 7-8 and 8-5.

The second term on the right-hand side of Equation (Equation 1) describes the valence interactions between farthest atoms in the cubenes; there are four longest bonds in each cubene; they are described by the harmonic potential:(3)$$\begin{array}{c} V_{k}=\frac{K}{2}\left(\right|r_{k}{\left|-\sqrt{3}\rho_{0}\right)}^{2}, \end{array}$$
where *K* is the bond stiffness the same as for the bonds between nearest atoms, $|r_{k}|$ is the distance between two farthest neighbors in a cubene molecule and $\sqrt{3}\rho_{0}=2.456$ Å is the equilibrium bond length. From Figure 1, it can be seen that the four bonds between farthest neighbors connect the following pairs of atoms: 1-7, 2-8, 3-5 and 4-6.

The third term on the right-hand side of Equation (Equation 1) describes the van der Waals interactions between carbon atoms of different cubenes. The computational cell includes 8M atoms numbered by the index i=1,2,...,8M. The *i*-th atom has the radius vector $r_{i}=\left(r_{i,x},r_{i,y},r_{i,z}\right)$, and $R_{i,j}=r_{j}-r_{i}$. The van der Waals interactions are described by the Lennard–Jones potential
(4)$$\begin{array}{c} \phi \left(\xi \right)=4\epsilon \left[{\left(\frac{\sigma }{\xi }\right)}^{12}-{\left(\frac{\sigma }{\xi }\right)}^{6}\right], \end{array}$$
where ϵ=0.002757 eV and σ=3.393 Å [49]. The van der Waals interactions are taken into account with the cut-off radius so that each cubene interacts with 12 nearest, 6 next-nearest, and 24 next-next-nearest cubenes. Van der Waals interactions between atoms belonging to the same cubene are not taken into account.

A comment should be made on the choice of interatomic potentials used in this work. To date, a number of interatomic potentials have been developed for carbon materials, the most popular of which are REBO-I [50], REBO-II [51], C-EDIP [52], Tersoff [53], LCBOP-I [54], ReaxFFC-2013 [55] and AIREBO [56] potentials and recently developed machine-learning potential GAP-20 [57]. Each potential has been designed for specific purposes. In this study, we do not consider high pressures or temperatures, so the potential can be greatly simplified to speed up the simulation. The van der Waals bonds describing the interaction between cubenes are two orders of magnitude weaker than covalent bonds within a cubene. Therefore, for our study for covalent bonds, the simplest harmonic potential is sufficient.

A home-made code written in the C++ algorithmic language is developed for simulations. Components of the stress tensor $\sigma_{ij}$ are calculated as described, e.g., in [58]. The gradient method is used to minimize the potential energy of the crystal Equation (Equation 1) with respect to the coordinates of carbon atoms and with respect to the translation vectors of the computational cell. The relaxation is complete when the maximal force acting on atoms does not exceed $10^{-9}$ eV/Å and the value of the maximal stress component does not exceed $10^{-6}$ eV/Å${}^{3}$.

For the relaxed structure, the potential energy per atom and the elastic constants $C_{ij}$ are calculated as described, for example, in [58]. By doing so, the elastic constants in Hooke’s law
(5)$$\left[\begin{array}{c} \sigma_{xx}\\ \sigma_{yy}\\ \sigma_{zz}\\ \sigma_{xy}\\ \sigma_{xz}\\ \sigma_{yz} \end{array}\right]=\left[\begin{array}{cccccc} C_{11} & C_{12} & C_{13} & C_{14} & C_{15} & C_{16}\\ C_{21} & C_{22} & C_{23} & C_{24} & C_{25} & C_{26}\\ C_{31} & C_{32} & C_{33} & C_{34} & C_{35} & C_{36}\\ C_{41} & C_{42} & C_{43} & C_{44} & C_{45} & C_{46}\\ C_{51} & C_{52} & C_{53} & C_{54} & C_{55} & C_{56}\\ C_{61} & C_{62} & C_{63} & C_{64} & C_{65} & C_{66} \end{array}\right]\left[\begin{array}{c} \varepsilon_{xx}\\ \varepsilon_{yy}\\ \varepsilon_{zz}\\ \varepsilon_{xy}\\ \varepsilon_{xz}\\ \varepsilon_{yz} \end{array}\right],$$
are determined. Note that, for any crystal, $C_{ij}=C_{ji}$ and, consequently, the maximum number of elastic constants is 21, although there are fewer of them for crystals with high symmetry.

We also calculate the potential energy, stress components, and elastic constants of the cubene crystal as the functions of homogeneous strain. After applying the homogeneous deformation, the structure is relaxed with respect to atomic displacements in order to achieve practically zero forces acting on the atoms. Stress components and elastic constants are calculated for relaxed structure. Briefly, the research flow can be described as follows. First, equilibrium phases are found, then the stability of the phases with respect to uniform deformation is analyzed, and then properties of the phases, such as energy, stress–strain curves, and elastic constants, are calculated.

## 3. Results

Let us present the numerical results obtained for cubene crystals. First, two equilibrium phases will be described. Then, the stability of the phases with respect to uniform elastic deformation will be demonstrated. Finally, elastic constants and stress–strain curves will be presented for both equilibrium phases.

### 3.1. Two Equilibrium Phases

The structure relaxation was carried out for 50 initial configurations with randomly oriented cubenes having centers of gravity at the sites of the fcc lattice with the distance between nearest neighbors equal to 5.3 Å. In many cases, metastable structures with a relatively high potential energy have been obtained. In some cases, two equilibrium phases with a relatively low potential energy were found. One of them has cubic symmetry, and the other belongs to the triclinic system. These two phases are described below.

The structure of the crystal will be defined by the translation vectors of the primitive cell, a,b,c, and coordinates of the carbon atoms in the cell, $\xi_{i}$. Then, the radius-vector of any atom in the crystal can be expressed as follows:(6)$$r_{\alpha \beta \gamma i}=\alpha a+\beta b+\gamma c+\xi_{i},$$
where α, β, and γ are arbitrary integers.

#### 3.1.1. Metastable Cubic Phase

A primitive translation cell of cubic phase includes two cubenes, see Figure 2. Translation vectors of a primitive cell for the cubic phase are
(7)a=(δ,0,0),b=(δ,δ,0),c=(δ/2,0,δ/2),
where δ=7.39223 Å. The translation vectors have lengths (in angstrom)
(8)$$\left|a\right|=\delta =7.392,\;|b|=\sqrt{2}\delta =10.454,\;\left|c\right|=\delta /\sqrt{2}=5.227.$$
The angles between translation vectors are
(9)$$\hat{a,b}=0.7854=45^{\circ },\;\hat{a,c}=0.7854=45^{\circ },\;\hat{b,c}=1.0472=60^{\circ }.$$
Volume per one cubene is 101.00 Å${}^{3}$.

Coordinates of 16 carbon atoms in a primitive translation cell are (in angstrom):

1-st cubene:

$\xi_{1}=\left(1.41966,0.00162041,1.4202\right)$,

$\xi_{2}=\left(1.89262,0.941731,0.470774\right)$,

$\xi_{3}=\left(0.942768,0.469339,-0.469764\right)$,

$\xi_{4}=\left(0.470234,-0.471199,0.480089\right)$,

$\xi_{5}=\left(0.479549,0.951046,1.89302\right)$,

$\xi_{6}=\left(0.952083,1.89158,0.943166\right)$,

$\xi_{7}=\left(0.0026573,1.41876,0.0030558\right)$,

$\xi_{8}=\left(-0.470304,0.478654,0.952482\right)$.

2-nd cubene:

$\xi_{9}=\left(5.11614,5.11488,0.00305583\right)$,

$\xi_{10}=\left(4.16686,5.5877,0.943167\right)$,

$\xi_{11}=\left(3.22632,4.63785,0.470774\right)$,

$\xi_{12}=\left(4.17603,4.16545,-0.469764\right)$,

$\xi_{13}=\left(5.58896,4.17477,0.952481\right)$,

$\xi_{14}=\left(4.63925,4.64716,1.89302\right)$,

$\xi_{15}=\left(3.69914,3.69774,1.4202\right)$,

$\xi_{16}=\left(4.64843,3.22492,0.480089\right)$.

#### 3.1.2. Stable Triclinic Phase

A primitive translation cell of triclinic phase includes one cubene, see Figure 3. Translation vectors of a primitive cell for the triclinic phase are (in angstrom)
(10)a=(7.34266,0,0),b=(3.34246,3.65657,0),c=(4.00020,0.359778,3.63883).
The translation vectors have lengths (in angstrom)
(11)|a|=7.343,|b|=4.954,|c|=5.420.
The angles between translation vectors are
(12)$$\hat{a,b}=0.8302=47.57^{\circ },\;\hat{a,c}=0.7405=42.43^{\circ },\;\hat{b,c}=0.9920=56.84^{\circ }.$$
The volume per cubene is 97.70 Å${}^{3}$, which is less than that of the cubic phase and, therefore, the triclinic phase is denser.

Coordinates of eight carbon atoms in a primitive translation cell are (in angstrom):

$\xi_{1}=\left(0.985844,-0.408445,0.510329\right)$,

$\xi_{2}=\left(1.907106,0.599395,0.891091\right)$,

$\xi_{3}=\left(1.348457,1.474166,-0.074438\right)$,

$\xi_{4}=\left(0.427158,0.466286,-0.455156\right)$,

$\xi_{5}=\left(0.064547,0.069598,1.475858\right)$,

$\xi_{6}=\left(0.985846,1.077477,1.856576\right)$,

$\xi_{7}=\left(0.427160,1.952208,0.891091\right)$,

$\xi_{8}=\left(-0.494102,0.944369,0.510329\right)$.

### 3.2. Energy of Cubic and Triclinic Cubene Crystals

Potential energy per atom for equilibrium cubic cubene is −0.045185 eV and, for equilibrium triclinic cubene, it is lower and equal to −0.047987 eV. The triclinic phase has the lowest energy among all structures obtained by relaxation of randomly oriented cubenes. We assume that the triclinic cubene crystal has the lowest possible energy, but we cannot provide a rigorous proof of this fact.

It is important to demonstrate that the crystals are stable with respect to application of elastic strain. The potential energy per atom was calculated for various values of the strain component $\varepsilon_{ij}$, while all other strain components were equal to zero. The results are plotted in Figure 4 for (a) cubic and (b) triclinic crystals. Note that the application of any strain component, positive or negative, leads to an increase in the potential energy of the crystal. This means that cubic and triclinic crystals are structurally stable up to ±5% strain.

For a cubic crystal, the curves for the normal strain components $\varepsilon_{xx}$, $\varepsilon_{yy}$, and $\varepsilon_{zz}$ in Figure 4a coincide, the same is true for the shear strains $\varepsilon_{xy}$, $\varepsilon_{xz}$, and $\varepsilon_{yz}$. This fact reflects the lattice symmetry.

For the low-symmetry triclinic phase, the curves in Figure 4b do not match, but the difference is small for normal and shear strains. It can be concluded that the triclinic phase is weakly anisotropic.

### 3.3. Elastic Constants for the Cubene Crystals

The stress components are related to the strain components according to Hooke’s law Equation (Equation 5), which for the cubic crystal assumes the form (elastic constants are given in GPa)
(13)$$\left[\begin{array}{c} \sigma_{xx}\\ \sigma_{yy}\\ \sigma_{zz}\\ \sigma_{xy}\\ \sigma_{xz}\\ \sigma_{yz} \end{array}\right]=\left[\begin{array}{cccccc} 464.5 & 177.1 & 177.1 & 0 & 0 & 0\\ 177.1 & 464.5 & 177.1 & 0 & 0 & 0\\ 177.1 & 177.1 & 464.5 & 0 & 0 & 0\\ 0 & 0 & 0 & 177.1 & 0 & 0\\ 0 & 0 & 0 & 0 & 177.1 & 0\\ 0 & 0 & 0 & 0 & 0 & 177.1 \end{array}\right]\left[\begin{array}{c} \varepsilon_{xx}\\ \varepsilon_{yy}\\ \varepsilon_{zz}\\ \varepsilon_{xy}\\ \varepsilon_{xz}\\ \varepsilon_{yz} \end{array}\right].$$

It is well known that isotropic elastic media are characterized by two independent elastic constants, while crystals with cubic symmetry generally have three independent elastic constants [59,60,61,62]. There are only two independent elastic constants in Hooke’s law Equation (Equation 13), since pairwise interatomic potentials are used, and the Cauchy relation is satisfied, which imposes an additional restriction on elastic constants [63].

For the triclinic cubene crystal, the Hooke’s law reads (elastic constants are given in GPa)
(14)$$\left[\begin{array}{c} \sigma_{xx}\\ \sigma_{yy}\\ \sigma_{zz}\\ \sigma_{xy}\\ \sigma_{xz}\\ \sigma_{yz} \end{array}\right]=\left[\begin{array}{cccccc} 477.0 & 187.0 & 183.8 & 9.0 & -9.9 & 16.2\\ 187.0 & 481.6 & 179.1 & 16.4 & 20.1 & -5.0\\ 183.8 & 179.1 & 484.6 & -26.0 & -9.6 & -10.5\\ 9.0 & 16.4 & -26.0 & 187.0 & 16.2 & 20.1\\ -9.9 & 20.1 & -9.6 & 16.2 & 183.8 & -26.0\\ 16.2 & -5.0 & -10.5 & 20.1 & -26.0 & 179.1 \end{array}\right]\left[\begin{array}{c} \varepsilon_{xx}\\ \varepsilon_{yy}\\ \varepsilon_{zz}\\ \varepsilon_{xy}\\ \varepsilon_{xz}\\ \varepsilon_{yz} \end{array}\right].$$

In Equation (Equation 14), due to the Cauchy relations, one has $C_{44}=C_{12}$, $C_{55}=C_{13}$, and $C_{66}=C_{23}$. Analysis of the elastic constants confirms that the triclinic cubene crystal is weakly anisotropic. The constants $C_{11}$, $C_{22}$, and $C_{33}$ differ by no more than 1.6%. The constants $C_{44}$, $C_{55}$, and $C_{66}$ differ by no more than 4.2%.

A comparison of the elastic constants of the cubic and triclinic cubenes indicates that the latter is stiffer. Indeed, in Equation (Equation 13), $C_{11}=C_{22}=C_{33}=464.5$ GPa, and, in Equation (Equation 14), these elastic constants are higher. The same applies to the shear moduli, which, for a cubic crystal are $C_{44}=C_{55}=C_{66}=177.1$ GPa, which is less than for a triclinic crystal. The higher stiffness of the triclinic crystal is due to its greater density.

### 3.4. Stress–Strain Curves

The stress–strain curves for cubic and triclinic cubene crystals are presented in Figure 5 and Figure 6, respectively. For the abscissa, the only nonzero strain component is used, and the other strain components are zero. The normal stress components are shown by black ($\sigma_{xx}$), red ($\sigma_{yy}$), and blue ($\sigma_{zz}$) lines. The shear stresses are plotted by yellow ($\sigma_{xy}$), cyan ($\sigma_{xz}$), and gray ($\sigma_{yz}$) lines. In some cases, dashed lines are used to show overlapping curves.

As can be seen from Figure 5 and Figure 6, when normal strain is applied, the shear stresses are equal to zero in cubic crystal, but, in the triclinic crystal, they are not.

For the normal stress components, as the functions of the normal strain components [panels (a) to (c) in Figure 5 and Figure 6] within the studied strain range ±5% demonstrate, there is a noticeable deviation from the linear dependence. On the other hand, the stress–strain curves $\sigma_{ij}\left(\varepsilon_{ij}\right)$ for i≠j show a linear relationship in this strain range (see panels (d) to (f) in Figure 5 and Figure 6).

The elastic constants $C_{11}$, $C_{22}$, and $C_{33}$ as the functions of strain are plotted in Figure 7 and Figure 8 for cubic and triclinic cubene crystals, respectively. In the studied strain range ±5%, these elastic constants depend almost linearly on the normal strain components and weakly depend on the shear strain components.

The elastic constants $C_{44}$, $C_{55}$, and $C_{66}$ as the functions of strain are plotted in Figure 9 and Figure 10 for cubic and triclinic cubene crystals, respectively. In the studied strain range ±5%, these elastic constants depend almost linearly on the normal strain components and weakly depend on the shear strain components.

## 4. Discussion

Two equilibrium phases of the C${}_{8}$ molecular crystal were found for the first time based on perturbative relaxation molecular dynamics simulations. The phase of cubic symmetry is metastable with a sublimation energy of −0.045 eV per atom. A primitive translational cell of a cubic cubene contains two cubenes with the volume of 101 Å${}^{3}$ per one cubene, see Figure 2. A triclinic cubene has a sublimation energy of −0.048 eV per atom, its primitive translational cell includes one cubene, and each cubene occupies a volume of 97.7 Å${}^{3}$, see Figure 3. The triclinic phase is most likely the ground state of the cubene crystal.

Stress–strain curves for cubic and triclinic cubenes were calculated in the strain range ±5%, see Figure 5 and Figure 6, respectively. No phase transformations were observed in this strain range; therefore, both phases in this strain range are structurally stable. Components of normal stress in the strain range ±5% show noticeable deviation from the linear Hooke’s law.

The elastic constants $C_{11}$, $C_{22}$, and $C_{33}$ as the functions of strain for cubic and triclinic cubene crystals are presented in Figure 7 and Figure 8, respectively. Shear moduli $C_{44}$, $C_{55}$, and $C_{66}$ depending on strain for cubic and triclinic cubene crystals are shown in Figure 9 and Figure 10, respectively. Elastic constants $C_{11}$, $C_{22}$, and $C_{33}$ depend almost linearly on strain, while constants $C_{44}$, $C_{55}$, and $C_{66}$ are weakly dependent on strain.

Elastic properties of cubenes can be compared to that of fullerenes. Young’s modulus of the fcc single crystal fullerite C${}_{60}$ at room temperature was measured in [64] to be 20±5 GPa. For C${}_{60}$ fullerene nanowires, the Young’s modulus is 15.1±1.39 [65], while, in bcc tetragonal C${}_{60}$ nanowire, it is 30 GPa [66]. According to our calculations, stiffness of cubenes is one order of magnitude higher, see Equations (Equation 13) and (Equation 14). This difference is related to the small size of C${}_{8}$ cubenes, which greatly increases the number of van der Waals bonds per atom. In addition, note that our calculations are made for zero temperature and, as is well known, the rigidity of molecular crystals decreases very rapidly with increasing temperature [64].

The limitations of our study are as follows. Since the harmonic potential is used to describe covalent bonds, high pressure and the possible transformation of the structure due to polymerization cannot be modeled within our model. The influence of temperature was not taken into account and, consequently, the range of thermal stability of the obtained structures was not established.

## 5. Conclusions and Future Work

Two equilibrium phases of C${}_{8}$ fullerite crystal were discovered by the relaxational molecular dynamics method, one of which has cubic and the other triclinic symmetry, see Figure 2 and Figure 3, respectively. The energy per atom for the triclinic phase is 0.0028 eV lower than that for the cubic phase, and triclinic fullerite is presumably the ground state structure. Both phases are stable with respect to uniform strain ±5%, see Figure 4. The stability of structures with respect to thermal vibrations was not analyzed in this work.

Stress–strain curves for cubic and triclinic fullerite demonstrate noticeable nonlinearity in the range of uniform normal strain ±5%, see panels (a)–(c) in Figure 5 and Figure 6. On the other hand, stress–strain curves for shear strain are almost linear, see panels (d)–(f) in Figure 5 and Figure 6.

The elastic constants $C_{11}$, $C_{22}$, and $C_{33}$ as the functions of strain are presented in Figure 7 and Figure 8 for cubic and triclinic cubenes, respectively. Similar results for shear moduli $C_{44}$, $C_{55}$, and $C_{66}$ can be seen in Figure 9 and Figure 10 for cubic and triclinic cubenes, respectively. Elastic constants $C_{11}$, $C_{22}$, and $C_{33}$ vary almost linearly with strain, while constants $C_{44}$, $C_{55}$, and $C_{66}$ are weakly dependent on strain.

The results presented here are the first step in studying the physical properties of cubenes. Future studies will analyze phonon dispersion curves, nonlinear dynamics of cubenes including rotobreathers [31,32,33,34,35], and crowdions [67]. To solve these problems, a more complex interatomic potential should be used [50,51,52,53,54,55,56,57]. The effect of temperature on the mechanical properties and thermal expansion will be analyzed. Another interesting line of future research is the development of continuum models capable of describing the vibrational spectra of cubenes. We hope that our study will stimulate experimental work on the mechanical and physical properties of cubenes. In particular, it would be of interest to analyze the polymerization of cubenes during high-temperature treatment under high pressure, similar to what was done in [68] for C${}_{60}$ and C${}_{70}$ fullerenes.

## Figures and Tables

**Figure 1 materials-15-04871-f001:**
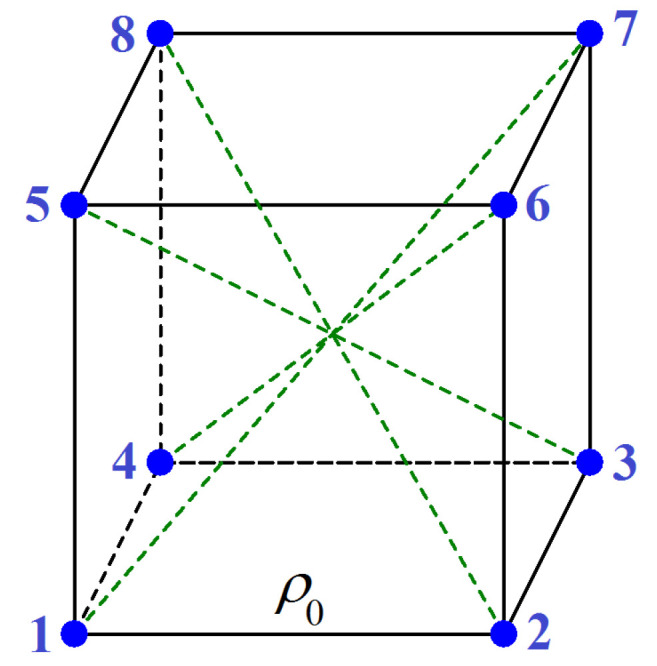
Cubene molecule model. Each vertex of a cube with side $\rho_{0}$ contains a carbon atom. Numbering of atoms in a cubene is shown. It is assumed that each carbon atom interacts through covalent bonds with the three nearest atoms and with the farthest atom; for example, atom 1 interacts with the atoms 2, 4, 5 and 7. Covalent bonds are described by the harmonic potential of stiffness *K* and equilibrium length $\rho_{0}$ for the nearest neighbors and $\sqrt{3}\rho_{0}$ for the farthest neighbors.

**Figure 2 materials-15-04871-f002:**
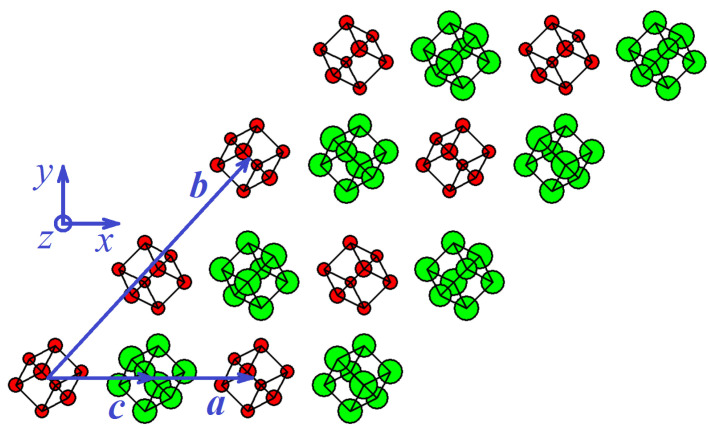
Structure of cubic cubene. The larger the *z* coordinate, the larger circle shows the carbon atom. Red (green) color is used for the lower (upper) plane of the structure (only two planes of cubenes are shown). The centers of gravity of the cubenes are close to the points of the fcc lattice. Vectors of the primitive translation cell, a, b, and c, are given by Equation (Equation 7). A primitive translational cell includes two cubenes differently oriented in space.

**Figure 3 materials-15-04871-f003:**
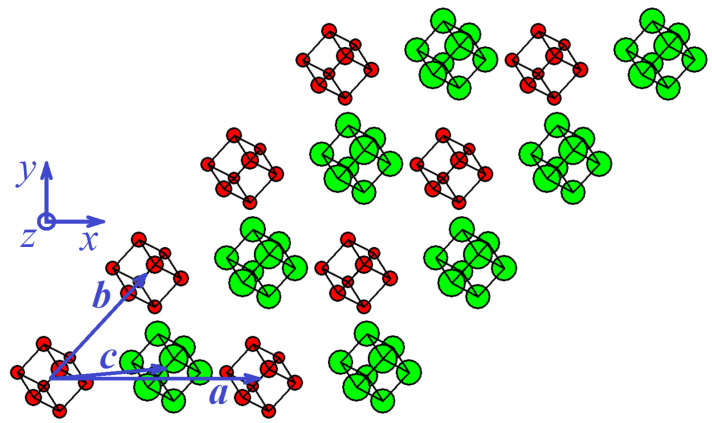
Structure of triclinic cubene. The larger the *z* coordinate, the larger circle shows the carbon atom. Red (green) color is used for the lower (upper) plane of the structure (only two planes of cubenes are shown). Vectors of the primitive translation cell, a, b, and c, are given by Equation (Equation 10). A primitive translational cell includes one cubene; therefore, all cubenes have the same orientation in space.

**Figure 4 materials-15-04871-f004:**
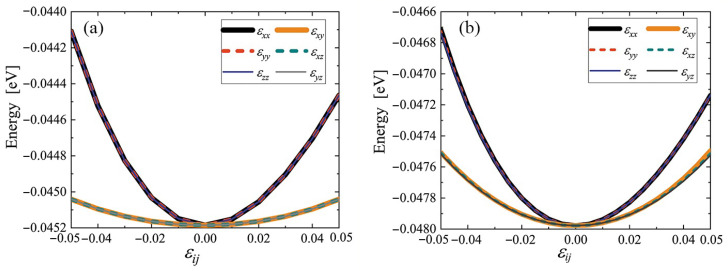
Potential energy per atom of cubic (**a**) and triclinic (**b**) cubene crystals depending on strain $\varepsilon_{ij}$. Only one component of the strain tensor is varied, the rest are equal to zero.

**Figure 5 materials-15-04871-f005:**
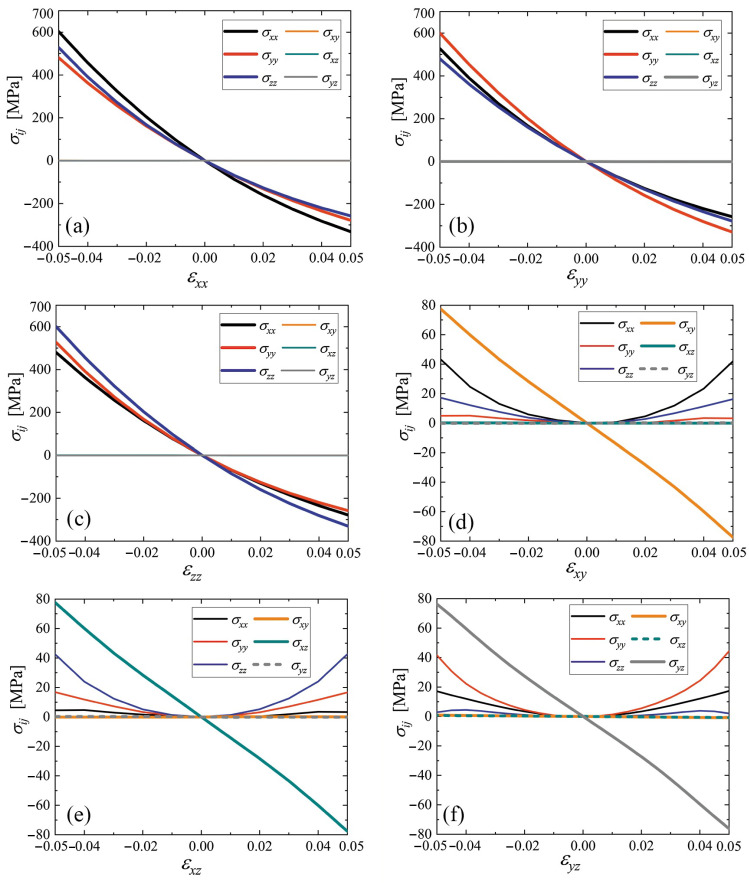
Stress–strain curves for cubic cubene crystal. Only one component of the strain tensor is varied, and the rest are equal to zero. The normal stress components are shown by black ($\sigma_{xx}$), red ($\sigma_{yy}$), and blue ($\sigma_{zz}$) lines. The shear stresses are plotted by yellow ($\sigma_{xy}$), cyan ($\sigma_{xz}$), and gray ($\sigma_{yz}$) lines. In some cases, dashed lines are used to show overlapping curves. Panels (**a**–**f**) correspond to changing $\varepsilon_{xx}$ to $\varepsilon_{yz}$, respectively.

**Figure 6 materials-15-04871-f006:**
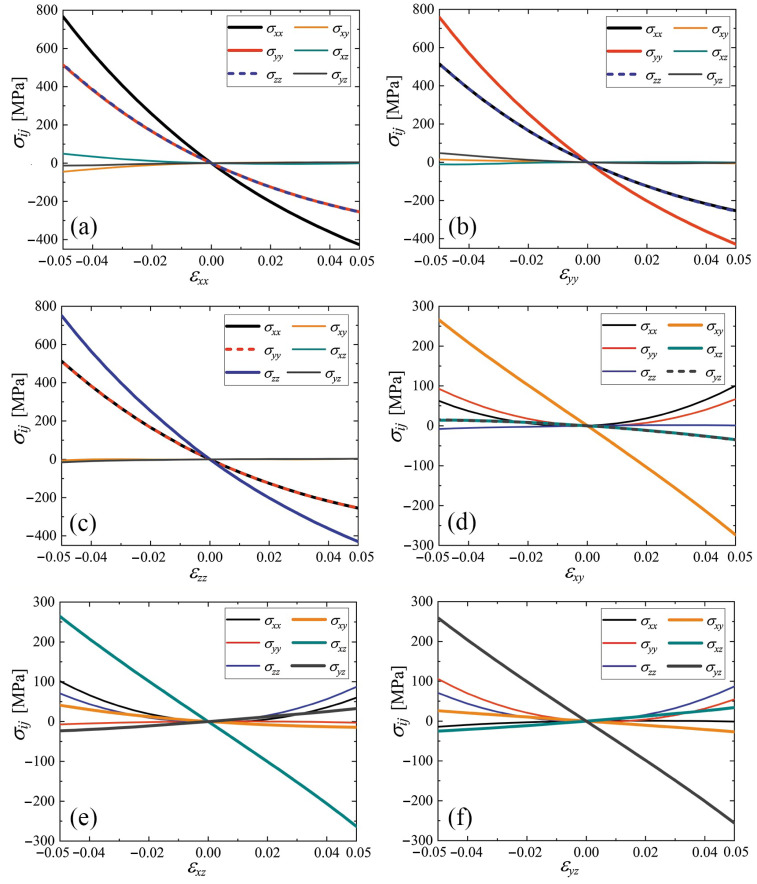
Stress–strain curves for triclinic cubene crystal. Only one component of the strain tensor is varied, the rest are equal to zero. Line colors are the same as in Figure 5. Panels (**a**–**f**) correspond to changing $\varepsilon_{xx}$ to $\varepsilon_{yz}$, respectively.

**Figure 7 materials-15-04871-f007:**
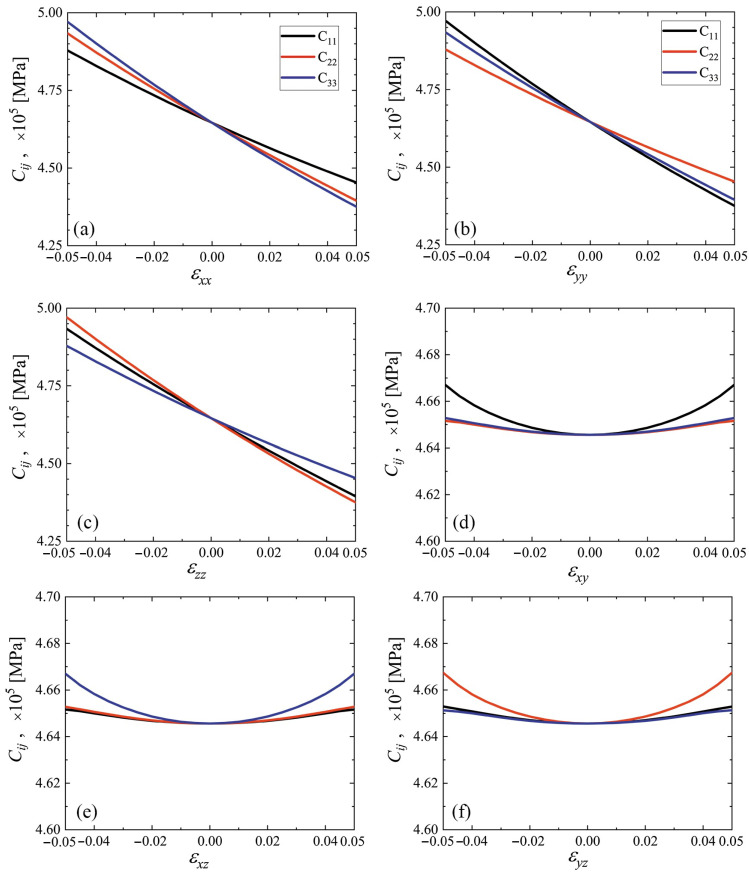
Elastic constants $C_{11}$, $C_{22}$, and $C_{33}$ as the functions of strain for cubic cubene crystal (black, red, and blue lines, respectively). Only one component of the strain tensor is varied, the rest are equal to zero. Panels (**a**–**f**) correspond to changing $\varepsilon_{xx}$ to $\varepsilon_{yz}$, respectively.

**Figure 8 materials-15-04871-f008:**
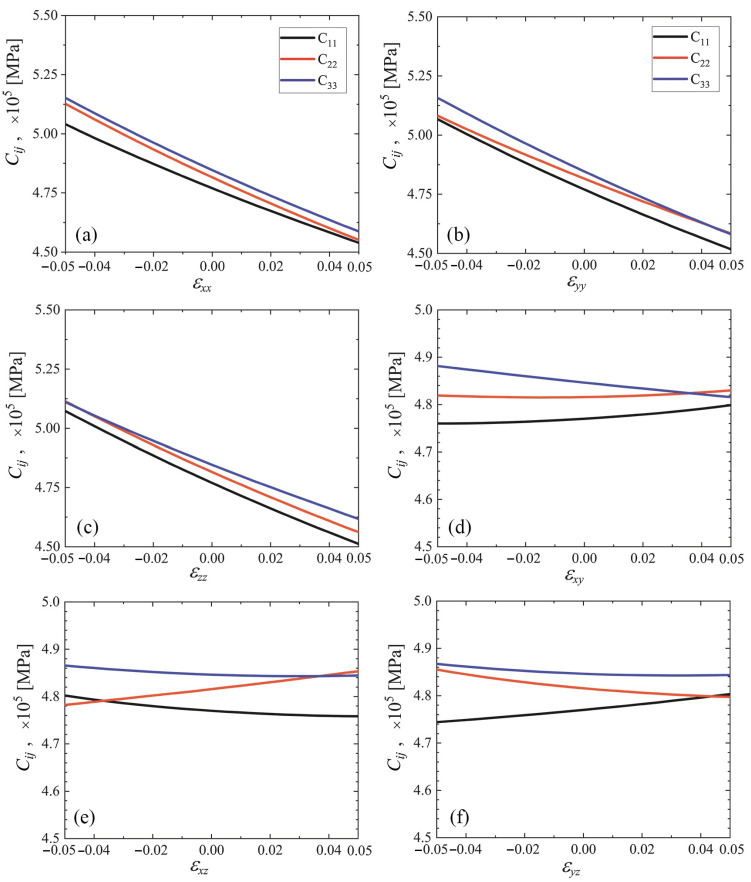
The same as in Figure 7, but for the triclinic crystal. Panels (**a**–**f**) correspond to changing $\varepsilon_{xx}$ to $\varepsilon_{yz}$, respectively.

**Figure 9 materials-15-04871-f009:**
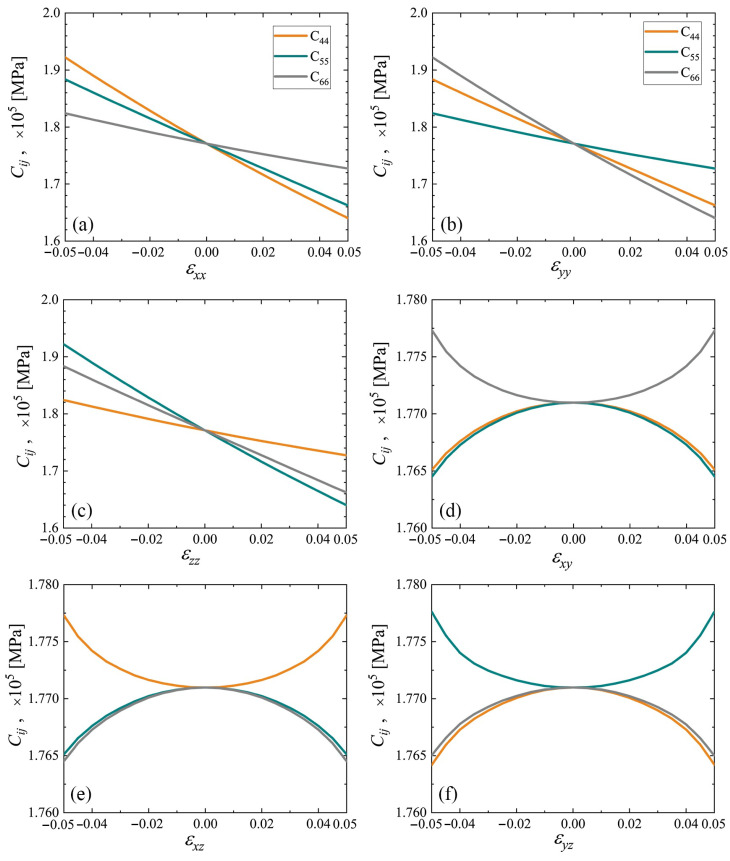
Elastic constants $C_{44}$, $C_{55}$, and $C_{66}$ as the functions of strain for cubic cubene crystal (yellow, cyan, and gray lines, respectively). Only one component of the strain tensor is varied, the rest are equal to zero. Panels (**a**–**f**) correspond to changing $\varepsilon_{xx}$ to $\varepsilon_{yz}$, respectively.

**Figure 10 materials-15-04871-f010:**
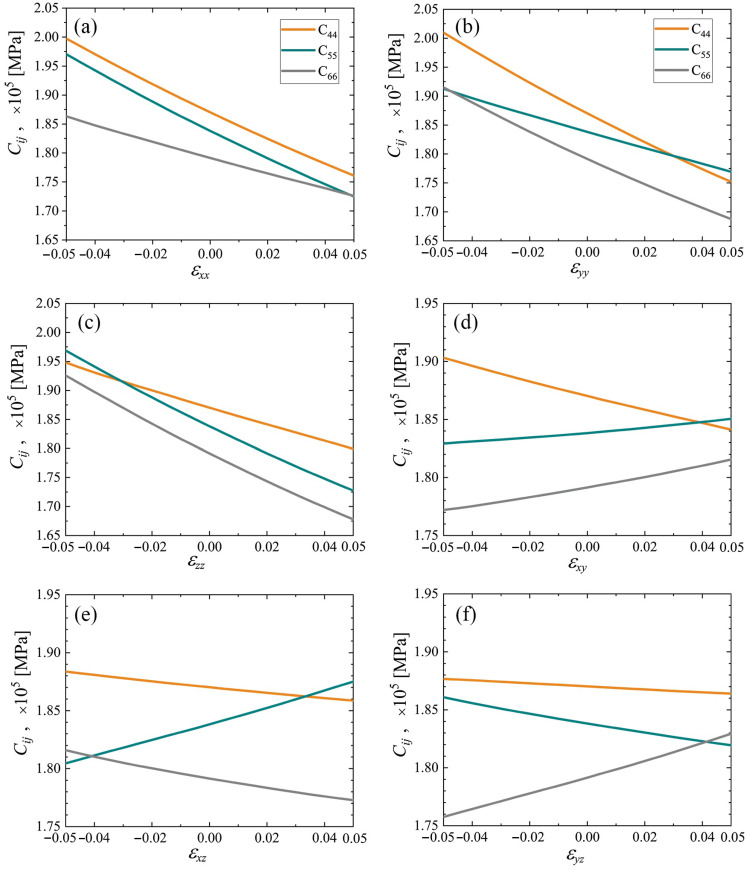
The same as in Figure 9, but for the triclinic crystal. Panels (**a**–**f**) correspond to changing $\varepsilon_{xx}$ to $\varepsilon_{yz}$, respectively.

## Data Availability

The data of this study are available from the corresponding author upon reasonable request.

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
