# Peer review of "Mechanical Properties of Cubene Crystals"

_materials, 2022, doi:10.3390/ma15144871_

Round 1

Reviewer 1 Report

Authors have conducted MD studies on Cubene crystals. The results are interesting. Some suggestions before the manuscript can be submitted in present form:

1. In abstract section, authors indicate "relaxational molecular dynamics". However there has been no further discussion on this term in the manuscript text.

2. What is the reference source for the potential for describing the state of cubene crystal as defined in Eqn. 1?

3. Why did not the authors consider Tersoff or REBO potential which describes the short term and long term forces in C-C bond? This should be described in the manuscript.

4. The conclusion section should be brief and should highlight the main contributions of the work.

Author Response

Reviewer #1:

Authors have conducted MD studies on Cubene crystals. The results are interesting. Some suggestions before the manuscript can be submitted in present form:

  1. In abstract section, authors indicate "relaxational molecular dynamics". However there has been no further discussion on this term in the manuscript text.

Our response:

The following text was added to Section 2. Materials and Methods:

“The equilibrium phases of cubene crystal are found using perturbative relaxational molecular dynamics. In this approach, the positions of the atoms are first perturbed to drive the system away from a local potential minimum, and then the structure is relaxed to reach a deeper energy minimum.”

  1. What is the reference source for the potential for describing the state of cubene crystal as defined in Eqn. 1?

Our response:

The reference was added as follows:

“Our model is built in the spirit of the work [20], where the deformation of valence bonds is described by the harmonic potential, and intermolecular interactions - by the Lennard-Jones potential.”

  1. Why did not the authors consider Tersoff or REBO potential which describes the short term and long term forces in C-C bond? This should be described in the manuscript.

Our response:

The following discussion was added to the Section 2. Materials and Methods:

“A comment should be made on the choice of interatomic potentials used in this work. To date, a number of interatomic potentials have been developed for carbon materials, the most popular of which are REBO-I [50], REBO-II [51], C-EDIP [52], Tersoff [53], LCBOP-I [54], ReaxFFC-2013 [55] and AIREBO [56] potentials and recently developed machine-learning potential GAP-20 [57]. Each potential has been designed for specific purposes. In this study, we do not consider high pressures or temperatures, so the potential can be greatly simplified to speed up the simulation. The van der Waals bonds describing the interaction between cubenes are two orders of magnitude weaker than covalent bonds within a cubene. Therefore, for our study for covalent bonds, the simplest harmonic potential is sufficient.”

  1. The conclusion section should be brief and should highlight the main contributions of the work.

We have separated the Discussion and Conclusion sections. This allowed us to briefly present the findings and highlight the main contributions of the work.

Conclusions now read:

Two equilibrium phases of C$_8$ fullerite crystal were discovered by the relaxational molecular dynamics method, one of which has cubic and the other triclinic symmetry, see Fig. 2 and 3, respectively. The energy per atom for the triclinic phase is 0.0028 eV lower than that for the cubic phase, and triclinic fullerite is presumably the ground state structure. Both phases are stable with respect to uniform strain ±5%, see Figure 4. The stability of structures with respect to thermal vibrations was not analyzed in this work.

Stress-strain curves for cubic and triclinic fullerite demonstrate noticeable nonlinearity in the range of uniform normal strain ±5%, see panels (a)-(c) in Figures 5 and 6. On the other hand, stress-strain curves for shear strain are almost linear, see panels (d)-(f) in Figures 5 and 6.

The stiffness constants $C_{11}$, $C_{22}$, and $C_{33}$ as the functions of strain are presented in Figures 7 and 8 for cubic and triclinic cubenes, respectively. Similar results for shear moduli $C_{44}$, $C_{55}$, and $C_{66}$ can be seen in Figures 9 and 10 for cubic and triclinic cubenes, respectively. Stiffness constants $C_{11}$, $C_{22}$, and $C_{33}$ vary almost linearly with strain, while constants $C_{44}$, $C_{55}$, and $C_{66}$ are weakly dependent on strain.

Reviewer 2 Report

In this paper, two cubene-based fullerenes, one of which belongs to the cubic system and the other to the triclinic system, are described based on relaxation molecular dynamics. The potential energy per atom, stiffness constants, and mechanical stress components as functions of lattice strain were calculated. It was determined that the cubene crystal is metastable, while the triclinic crystal is a putative ground-state crystalline phase. The results presented here represent a first step in the study of the physical and mechanical properties of C8 fullerite, which may have potential applications in hydrogen storage and other applications.

The paper is well written, easy to read and the theoretical results presented are very supportive of the experimental work, especially those concerning Cij coefficients. It is a great novelty for these carbon structures and the determination of the above indicated parameters. I fully recommend it for publication in Materials.

Author Response

We are pleased with the high appraisal of our work.

Reviewer 3 Report

1.      The uppercase and lowercase of the title should be corrected.

2.      Orcid ID does not type like presently, please delete it.

3.      The abstract should be added quantitative results.

4.      Keywords should reorder based on alphabetical order.

5.      Construct a paragraph with 3 sentences or more with one sentence of the main sentence followed by a supporting sentence. The authors sometimes made the paragraph only with 1-2 sentences. For example, in lines 56-57.

6.      The present study does not bring significant novelty. Nothing really now was shown in the existence of the present study. Similar literature has been published. It needs to be focused on by the authors. If this issue cannot be addressed by the authors, the current publication is should be rejected.

7.      Crystal has been potentially used in several applications, such as bearing material for total hip implants. This important thing should be informed in the introduction and/or discussion section. Also, to support this sentence it needed to adopt the suggested reference published by MDPI as follows: Computational Contact Pressure Prediction of CoCrMo, SS 316L and Ti6Al4V Femoral Head against UHMWPE Acetabular Cup under Gait Cycle. J. Funct. Biomater. 2022, 13, 64. https://doi.org/10.3390/jfb13020064

8.      The authors need to explain the previous research with their findings and its shortcoming. It will show the research gap infilled with present novelty. Please highlight it more advance.

9.      The research flow should be explained as an illustrative figure in the materials and methods section to make the reader more interested and easier to understand.

10.   Detailed information regarding tools/experimental setup should be explained in the materials and methods section.

11.   Accuracy and tolerance of the experimental setup should be stated in the materials and methods section.

12.   Basis/standard/regulation as fundamental information should be described in the materials and methods section.

13.   The materials and methods section needs further explanation in specific procedures since the present manuscript only explains the very brief procedure.

14.   Legends in Figures 5, 6, and 8 should only make it into 1 since the six legends are still the same.

15.   The discussion and conclusion sections should be separated.

16.   The limitations of the present study should be explained before the conclusion section.

17.   Further research needs to be explained in the abstract section.

18.   Institutional Review Board Statement, Informed Consent Statement, and Acknowledgment information should be given. If it is not, just type “Not applicable”.

19.   Please used Materials, MDPI format properly. The present form does not.

20.   The authors need to proofread the manuscript to enhance the English language. It is related to my comment number 5.

Author Response

Reviewer #3:

  1. The uppercase and lowercase of the title should be corrected.

Our response:

We have capitalized the first letters in the title according to the style of Materials.

  1. Orcid ID does not type like presently, please delete it.

Our response:

For some reason we cannot remove the Orcid ID using the Latex template provided by the journal. We hope they will be deleted by the Editors if the manuscript is accepted.

  1. The abstract should be added quantitative results.

Our response:

The following sentences were added to the abstract:

(the potential energies per atom for these two structures are -0.0452 and -0.0480~eV, respectively). The cubic phase has a lower density than the monoclinic one (volumes per cubene are 101 and 97.7A^3). The stiffness constants for the monoclinic phase are approximately 4\% higher than those for the cubic phase.

  1. Keywords should reorder based on alphabetical order.

Our response:

The keywords were arranged in alphabetical order.

  1. Construct a paragraph with 3 sentences or more with one sentence of the main sentence followed by a supporting sentence. The authors sometimes made the paragraph only with 1-2 sentences. For example, in lines 56-57.

Our response:

We have expanded short paragraphs. The following text was added:

Carbon-carbon interactions are described by the harmonic potential, while non-valence interactions are described by the Lennard-Jones potential. Each cubene interacts with the nearest, second and third neighboring cubenes. Below is a detailed description of the model.

First, two equilibrium phases will be described. Then the stability of the phases with respect to uniform elastic deformation will be demonstrated. Finally, elastic constants and stress-strain curves will be presented for both equilibrium phases.

  1. The present study does not bring significant novelty. Nothing really now was shown in the existence of the present study. Similar literature has been published. It needs to be focused on by the authors. If this issue cannot be addressed by the authors, the current publication is should be rejected.

Our response:

We emphasized the novelty of our work in the abstract and conclusions.

In the abstract:

“In this work, based on relaxational molecular dynamics, two fullerites based on cubenes are described for the first time,…”

In Conclusions:

“Two equilibrium phases of the C$_{8}$ molecular crystal were found for the first time…”

  1. Crystal has been potentially used in several applications, such as bearing material for total hip implants. This important thing should be informed in the introduction and/or discussion section. Also, to support this sentence it needed to adopt the suggested reference published by MDPI as follows: Computational Contact Pressure Prediction of CoCrMo, SS 316L and Ti6Al4V Femoral Head against UHMWPE Acetabular Cup under Gait Cycle. J. Funct. Biomater. 2022, 13, 64. https://doi.org/10.3390/jfb13020064

Our response:

The work was cited in the Introduction as follows:

The crystal has been potentially used in several applications, such as bearing material for total hip implants [43].

  1. The authors need to explain the previous research with their findings and its shortcoming. It will show the research gap infilled with present novelty. Please highlight it more advance.

Our response:

The following text was added to the Introduction in order to describe the previous research:

The crystal structure of fullerites is of great importance, since the structure determines the properties of crystals. Fullerite C$_{60}$ can be packed into a simple cubic or face-centered cubic (fcc) lattice [44,45]. In both lattices, a primitive translational cell contains one fullerene molecule. In the theoretical work [46] it was shown that for C$_{60}$ the fcc structure is more stable than hexagonal close-packing (hcp), but for C$_{70}$ hcp has lower energy than fcc lattice. The fcc structure of C$_{60}$ at 0 K has an orthorhombic distortion due to ordering of fullerenes. At higher temperatures there is a transition to an orientationally disordered fcc lattice [46]. As can be seen, many studies have been carried out to predict and analyze the structure of fullerites C$_{60}$ and C$_{70}$, but we have not been able to find any information about structure for fullerite C$_{8}$ except for the study [47] where cubic cubane C8H8 was described. In the present study, we set ourselves the task of finding the equilibrium phases of a cubene crystal and their mechanical properties.

  1. The research flow should be explained as an illustrative figure in the materials and methods section to make the reader more interested and easier to understand.

Our response:

We have described the research flow verbally as follows:

Briefly, the research flow can be described as follows. First, equilibrium phases are found, then the stability of the phases with respect to uniform deformation is analyzed, and then properties of the phases such as energy, stress-strain curves and stiffness constants are calculated.

  1. Detailed information regarding tools/experimental setup should be explained in the materials and methods section.

Our response:

In our theoretical study tools/ experimental setup are not relevant.

  1. Accuracy and tolerance of the experimental setup should be stated in the materials and methods section.

Our response:

We have specified the accuracy parameters used in our numerical experiments:

The relaxation is complete when the maximal force acting on atoms does not exceed 10^{-9} eV/A and the value of the maximal stress component does not exceed 10^{-6} eV/A^3.

  1. Basis/standard/regulation as fundamental information should be described in the materials and methods section.

Our response:

We believe that there are no regulations for the studies such as our work.

  1. The materials and methods section needs further explanation in specific procedures since the present manuscript only explains the very brief procedure.

Our response:

We have extended the motivation and the description of the interatomic potentials used in our study; this was also the request from the Reviewer #1. We have added more information to the section 2. Materials and Methods:

“A comment should be made on the choice of interatomic potentials used in this work. To date, a number of interatomic potentials have been developed for carbon materials, the most popular of which are REBO-I [50], REBO-II [51], C-EDIP [52], Tersoff [53], LCBOP-I [54], ReaxFFC-2013 [55] and AIREBO [56] potentials and recently developed machine-learning potential GAP-20 [57]. Each potential has been designed for specific purposes. In this study, we do not consider high pressures or temperatures, so the potential can be greatly simplified to speed up the simulation. The van der Waals bonds describing the interaction between cubenes are two orders of magnitude weaker than covalent bonds within a cubene. Therefore, for our study for covalent bonds, the simplest harmonic potential is sufficient.”

  1. Legends in Figures 5, 6, and 8 should only make it into 1 since the six legends are still the same.

Our response:

In Figures 5 and 6 we show the stress components as the functions of strain components for cubic and triclinic cubenes, respectively. Due to the symmetry of the lattice, some of the stress-strain curves coincide. To show two overlapping curves one of them is plotted by the solid and the other by the dashed line. The legends in Figures 5 and 6 show which line is used for which curve. In Figures 7 and 8, as well as in Figures 9 and 10, the curves do not overlap and we use legends only ones for each column of figure panels.

  1. The discussion and conclusion sections should be separated.

Our response:

We have separated the discussion and conclusion sections.

  1. The limitations of the present study should be explained before the conclusion section.

Our response:

The following text was added at the end of the Discussion section:

“The limitations of our study are as follows. Since the harmonic potential is used to describe covalent bonds, high pressure and the possible transformation of the structure due to polymerization cannot be modeled within our model. The influence of temperature was not taken into account and, consequently, the range of thermal stability of the obtained structures was not established.”

  1. Further research needs to be explained in the abstract section.

Our response:

The following sentence was added to the abstract:

“In the future, the influence of temperature on the properties of cubenes will be analyzed.”

  1. Institutional Review Board Statement, Informed Consent Statement, and Acknowledgment information should be given. If it is not, just type “Not applicable”.

Our response:

For our manuscript the applicable statements are the following:

Author Contributions: Conceptualization, S.V.D. and I.S.P.; methodology, I.S.P.; software, I.V.K.; investigation, L.Kh.R. and A.M.B.; writing - original draft preparation, S.V.D.; writing review and editing, S.V.D. and I.S.P. All authors have read and agreed to the published version of the manuscript. Funding: For I.S.P, A.M.B. and S.V.D. this research was funded by the Russian Science Foundation, grant no. 21-19-00813.

Funding: For I.S.P, A.M.B. and S.V.D. this research was funded by the Russian Science Foundation, grant no. 21-19-00813.

Data Availability: The data of this study are available from the corresponding author upon reasonable request.

Conflicts of Interest: The authors declare no conflict of interest.

  1. Please used Materials, MDPI format properly. The present form does not. 

Our response:

We used the MDPI Latex template and made some styling corrections in the revised version of the manuscript.

  1. The authors need to proofread the manuscript to enhance the English language. It is related to my comment number 5.

Our response:

We have done our best to proofread the manuscript and have made corrections related to comment number 5.

Round 2

Reviewer 3 Report

Good job for the authors.